# Transparent origami glass

Yang Xu[1], Ye Li[1], Ning Zheng [1✉], Qian Zhao [1,2] & Tao Xie [1,2✉]

The art of origami has emerged as an engineering tool with ever increasing potential, but the technique is typically limited to soft and deformable materials. Glass is indispensable in many applications, but its processing options are limited by its brittle nature and the requirement to achieve optical transparency. We report a strategy that allows making three dimensional transparent glass with origami techniques. Our process starts from a dynamic covalent polymer matrix with homogeneously dispersed silica nanoparticles. Particle cavitation and dynamic bond exchange offer two complementary plasticity mechanisms that allow the nanocomposite to be permanently folded into designable geometries. Further pyrolysis and sintering convert it into transparent three dimensional glass. Our method expands the scope of glass shaping and potentially opens up its utilities in unexplored territories.

[1] State Key Laboratory of Chemical Engineering, College of Chemical and Biological Engineering, Zhejiang University, Hangzhou, People's Republic of China. [2] ZJU-Hangzhou Global Scientific and Technological Innovation Center, Hangzhou, People's Republic of China. ✉email: zhengning@zju.edu.cn; taoxie@zju.edu.cn

Glass is indispensable in many applications due to its excellent optical transparency, abrasion resistance, and thermal and chemical stability. However, its processing options are very limited compared to polymers and metals. Conventional glass shaping operates under harsh conditions such as high temperature or chemical etching. Sol-gel chemistry allows defining glass shapes under milder conditions, but the geometric complexity is inherently limited by the molding technique involved. Another strategy utilizes silica-polymer composites as the precursor for glass making[1]. This permits low-temperature molding. Subsequent machining and sintering produce 3D glass. Without using molding, precursor composites can also be 3D printed[2–7]. This has recently surfaced as an attractive way to produce glass with complex shapes. Despite the emergence of 3D glass printing methods[2–7], their typical layer-by-layer nature raises several issues: printing speed, resolution, and surface roughness. In addition, many 3D geometries require the use of support during printing and its subsequent removal afterward can be very cumbersome. Two-photon techniques and micro-3D printing allow producing high-resolution structures with smooth surfaces, but compromise markedly the print size and productivity[8]. Localized laser processing allows glass shaping, but the method is limited to simple geometric manipulation such as bending[9]. Other methods for making 3D ceramics[10–12] have also emerged, but they cannot be adopted for making glass due to the additional requirement of optical transparency.

Origami is a versatile method to convert a planar paper sheet into three dimensional (3D) geometries. As an ancient art, it has been endued with new vitality in modern times. In particular, the extension from paper sheets into a diverse set of materials[13–15] has unleashed vast potential in many engineering areas including soft robotics[16], wearable electronics[17], aerospace structures[18], and medical devices[19]. Despite the versatility, its direct extension into glass shaping is prohibitive since typical glass is rigid and brittle. The glass precursor composites mentioned above are not designed to be deformable either. We hypothesize that, with the delicate molecular design of the precursor composite, it is possible to introduce mechanisms that make it deform in such a way that permits origami-shaping of transparent glass.

## Results

**Processing of 3D transparent origami glass.** Our process is illustrated in Fig. 1a. A polymer composite sheet is obtained by curing a silica nanoparticle-filled liquid precursor. After cutting into the desired shape, the composite sheet is folded using an origami technique, which involves manual folding at room temperature much like actual paper origami. Further pyrolysis and sintering remove the polymer binder and convert the 3D object into glass. While seemingly simple, two requirements should be met to enable such a process. First, the process demands the origami deformed sheet to maintain its shape during the subsequent high-temperature pyrolysis step. This is a challenge since polymers typically possess entropic elasticity, namely, they would naturally recover to their original undeformed geometry when heated above the thermal transition (glass transition or melting transition)[20]. Second, the composite should be foldable like a paper sheet. This requires that it possesses suitable mechanical properties (modulus and stretchability).

To address the first requirement, two mechanisms (Fig. 1b) can be potentially relevant in order to suppress the undesirable elasticity. The first is physical plasticity. For polymer nanocomposites, cavitation as the result of filler-matrix interfacial delamination is an effective toughening mechanism[21]. For our purpose, we hypothesize that cavitation at room temperature can induce non-recoverable deformation (i.e., plasticity). The second

is chemical plasticity. One can use a dynamic covalent network polymer as the matrix, yielding non-recoverable deformation via network rearrangement at elevated temperatures[20,22,23]. Specifically, we design a polyester network (Fig. 1b, c) in which the exchange reaction (transesterification) between the dangling hydroxyl groups and the esters can lead to permanent deformation. Following the above design principles, we succeed in producing a 3D-shaped feather made of transparent glass (Fig. 1d) with superior thermal stability (Fig. 1e).

**Characterizations of the silica-polymer nanocomposites.** Enabling the above process requires fine tuning of the polymer composites. Herein, silica nanoparticles (50 nm) are dispersed in a liquid mixture containing reactive polycaprolactone diacrylate and 4-hydroxybutylacrylate, along with a sintering aid (phenoxyethanol), a UV curing initiator, a transesterification catalyst, and a solvent (dimethylformamide). Polymer composites are obtained by photocuring followed by solvent removal, with the related chemistry shown in Supplementary Fig. 1. Two major factors affect the process. We first investigate the impact of the ratio between polycaprolactone diacrylate (molecular mass: 2000) and 4-hydroxybutylacrylate on the chemical plasticity. A series of polymers are synthesized and denoted as OHX, with X representing the mass percentage of 4-hydroxybutylacrylate to the overall polymer. Their chemical plasticity is reflected in the isostrain stress relaxation behaviors at 130 °C (Fig. 2a). From OH0 to OH30, the stress relaxation rate increases with the content of the hydroxyl groups. When the hydroxyl group further increases to 50%, the stress relaxation kinetics becomes slower. This is most likely due to the fact that the reduction in the number of ester bonds overtakes the increase of the hydroxyl groups. We choose OH30 for further investigation hereafter. Its stress relaxation at 130 °C for 2 h is 85%, corresponding to shape retention of 87% (Supplementary Fig. 2). Herein, shape retention reflects quantitively the permanent shape alteration. It is defined as $R_{ren} = 100\% \times (\mathcal{E}/\mathcal{E}_{load})$, with $\mathcal{E}_{load}$ and $\mathcal{E}$ denoting the initially applied strain and the permanently fixed strain (after the stress relaxation and force removal).

Introducing silica nanoparticles affects greatly the mechanical properties of the composites, which are designated as PX with X being the mass percentage of the silica nanoparticle over the total weight of the composite (including the sintering aid phenoxyethanol). Figure 2b shows that the Young's modulus increases with the silica content. The strain at break increases initially at a filler content below 29% due to the toughening effect of silica[24]. Further increase of the filler content to 38% yields significant decline in strain at break because of the reduction of the deformable polymer matrix. Without the nanoparticle, the polymer behaves elastically with shape retention of 0. Incorporation of nanoparticles yields room-temperature physical plasticity, with the shape retention improving with their content (Fig. 2c). In addition to the impact on mechanical properties, a high filler content favors the later step of glass making. Overall, P29 shows the best balanced properties in terms of strain of break, shape retention, and filler content. Without heating, it can be permanently folded much like a paper sheet. Critically, the folding persists even after prolonged heating during the later pyrolysis step at 600 °C. P29 is therefore used for further study. Figure 2d shows that its shape retention under physical plasticity improves with the strain, reaching a plateau value of 37% at a strain of 16%. The tensile shape retention of P29 under chemical plasticity is 91 ± 2% (Supplementary Fig. 3a). By comparison, the bending shape retention after physical plasticity is 55% ± 3%, while the bending shape retention after chemical plasticity is 92% ± 3% (Supplementary Fig. 3b).

To understand the mechanism behind the room temperature plasticity, the composite surface morphologies before and after

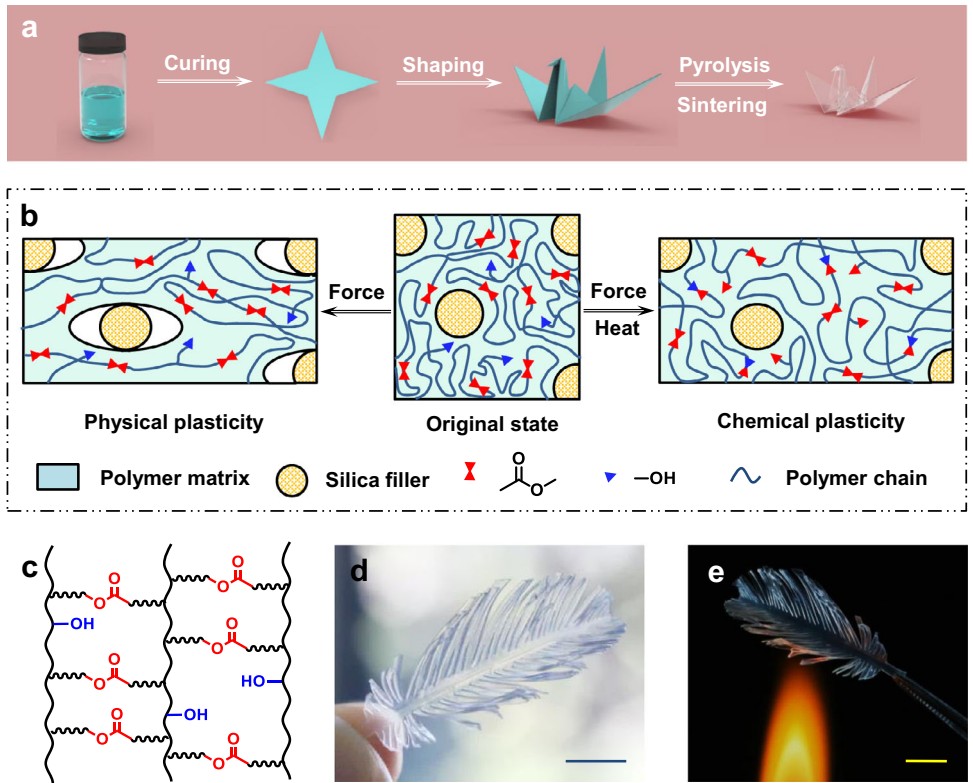

**Fig. 1 Fabrication of 3D transparent origami glass. a** Schematic illustration of the fabrication process. **b** Two mechanisms for permanent deformation via plasticity. **c** Dynamic polymer ester network with dangling hydroxyl groups. **d** Photographic image of a 3D transparent glass feather. Scale bar: 1 cm. **e** Demonstration of the high thermal resistance at 600 °C. Scale bar: 1 cm.

the stretching are analyzed. In contrast to the initial defect-free uniform distribution of silica (Fig. 2e), nano-cavities aligned in the loading direction (Fig. 2f) are formed as the result of the stretching. This observation verifies the cavitation induced physical plasticity in Fig. 1b. In addition to the physical plasticity, the chemical plasticity of the polymer matrix is also carried into the composite. At 130 °C, the nanocomposite can be permanently folded into a four-leaf clover. Importantly, this can be further folded into a windmill via a consecutive chemical plasticity step (Fig. 2g). This multi-step folding allows access to complex shapes by accumulating the shape changes from each individual step.

**Characterizations of the glass**. The 3D-shaped polymer composite can be turned into a transparent glass through pyrolysis and sintering following the heating procedure in Supplementary Fig. 4. In this process (Fig. 3a), the composite is first turned into a brown part (a porous intermediate) by pyrolyzing the organic scaffold. A subsequent vacuum sintering step turns the brown part into transparent glass. The linear shrinkage after sintering is 46% and the residual mass is around 29% (Supplementary Fig. 5), consistent with the nanofiller content. In contrast, the residual mass for the pure polymer network after pyrolysis is near zero (Supplementary Fig. 5), further implying that the glass is silica. The transmittance of the composite green body, the brown part, and the sintered glass are plotted in Fig. 3b, showing the large change during the process. The visible light transmission of the final glass is around 90%. Its surface roughness is below 17 nm (Supplementary Fig. 6), which is noteworthy since no polishing is involved. The photographic images of the brown part and transparent glass as well as their cross-section SEM images (Fig. 3c, d) are consistent with their optical measurement. Increasing the transmission beyond 90% is possible by further

optimization of the heating procedure and employing a higher vacuum during sintering. XRD analysis (Supplementary Fig. 7) confirms that the obtained glass is amorphous. The bending strength and hardness of the obtained glass, calculated from the mechanical curves in Supplementary Fig. 8, are 88.2 ± 5.3 MPa and 8.2 ± 0.3 GPa, respectively.

**Fabrication of 3D transparent origami glass**. We next proceed to combine the origami deformability of the composite green body with the glass making to produce transparent origami glass with complex 3D geometries. The overall process is illustrated in Fig. 4. The composite green body is first laser-cut into a desired pattern according to the designed two-dimensional model. Alternatively, it is also possible to use digital projector light to directly produce the flat cut pattern if a visible light initiator is employed. Next, the planar film is permanently deformed by origami technique. Specifically, the "crane" is made by physical plasticity whereas the "vase" and "flower" by chemical plasticity. The two plasticity mechanisms are complementary to each other. The physical plasticity is more convenient since it does not require heating during the folding. By comparison, chemical plasticity is more favored when high shape retention is needed. Nevertheless, the origami green bodies are turned into brown parts and transparent glass with high geometric fidelity in all the cases. In particular, no unpredictable deformation was observed during the pyrolysis and sintering process except the vase shape that requires an extra mechanical support. These origami glasses are challenging to produce with 3D printing as it would require extensive support structures that are cumbersome to remove. The surface smoothness is also difficult to achieve with regular 3D printing.

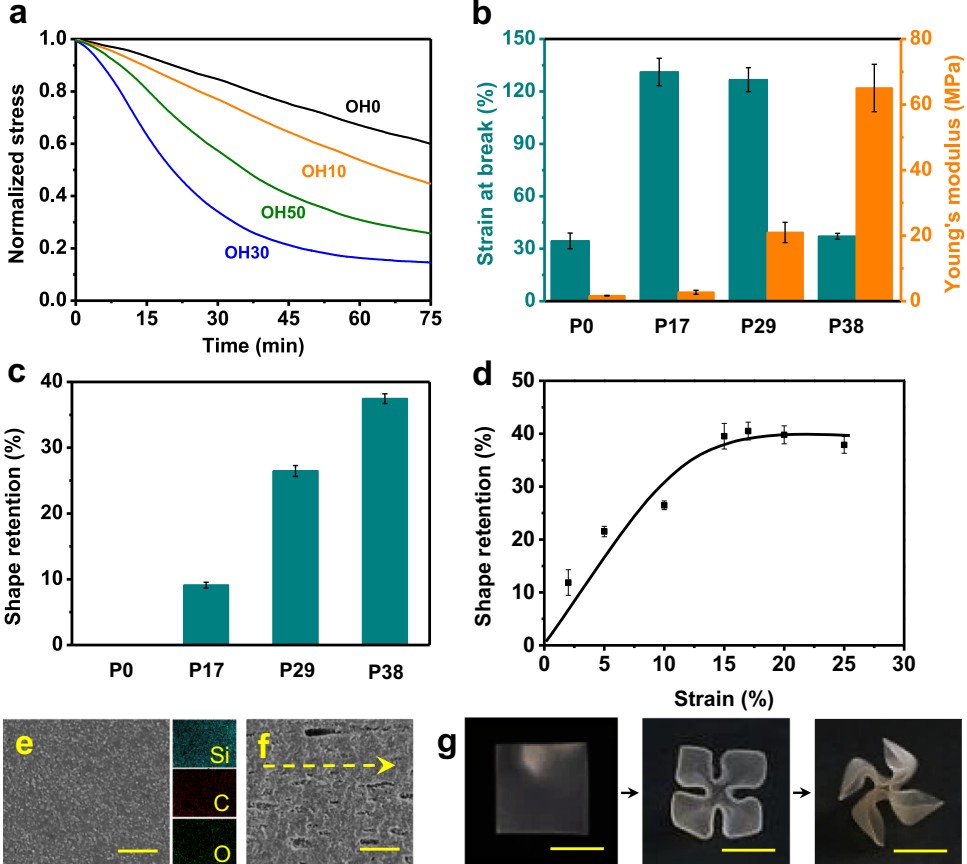

**Fig. 2 Characterizations of the silica-polymer nanocomposites. a** Isostrain stress relaxation of the neat polymer networks at 130 °C (strain is fixed at 10%). **b** Mechanical properties of composites with different contents of $SiO_2$ nanoparticles. **c** Shape retention of the composites measured for an initial strain of 10%. **d** The impact of strain on the shape retention for P29. **e** Scanning electron microscopic (SEM) image (left) and energy dispersive X-ray spectrometry (EDS) mapping (right) of the composite surface. Scale bar: 2 μm. **f** SEM image of the composite surface after stretching by 100%, with the arrow showing the stretching direction. Scale bar: 2 μm. **g** Multi-step accumulative shape reconfiguration of the composite, with each step occurring at 130 °C for 2 h. Scale bars: 1 cm. All error bars represent standard deviations.

## Discussion

Traditional glass making employs molding, blowing, and sometimes etching and polishing. In contrast, our origami glass making technique uses low power laser-cutting and folding. The laser-cutting is digitally controlled. The folding is manual, but can be automated with fixtures. Our technique is therefore suitable for large-scale manufacturing. More importantly, our process does not use molds, thus the geometric complexity is not bound by traditional molding techniques. Besides making attractive arts, it opens up future opportunities in making geometrically complex devices that require the use of glass as an essential component.

## Methods

**Materials**. Polycaprolactone diol ($M_n = 2000$) was purchased from Sigma Aldrich. 2-Isocyanatoethyl acrylate (stabilized with BHT), dibutyltin dilaurate (DBTDL), 4-hydroxybutyl acrylate (stabilized with MEHQ), and Irgacure 2959 were purchased from TCI. Silica nanoparticles (AEOSIL R972) were procured from Evonik. Phenoxyethanol (POE) was obtained from Aladdin. N,N-dimethylformamide (DMF), toluene, and methanol were purchased from Sinopharm. The solvents used were of analytical grade. All the chemicals were used as received.

**Synthesis and characterization of polycaprolactone diacrylate**. Polycaprolactone diol (30 g) was melted at 120 °C in a flask. Toluene (150 g), 2-isocyanatoethyl acrylate (6.35 g), and dibutyltin dilaurate (0.15 g, the reaction catalyst) were subsequently added. The obtained solution was allowed to react for 6 h at 80 °C with magnetic stirring. The resultant polycaprolactone diacrylate was obtained by precipitating in methanol. The precipitate was then vacuum dried at room temperature for 24 h. We note that 2-isocyanatoethyl acrylate was used in excess to ensure high conversion of the hydroxyl

groups and the unreacted 2-isocyanatoethyl acrylate monomer was removed in the subsequent purification process.

**Preparation of the polymer matrixes**. For OH30, polycaprolactone diacrylate (10 g), 4-hydroxybutyl acrylate (3 g), Irgacure 2959 as the UV initiator (0.13 g), dibutyltin dilaurate (0.13 g, the transesterification catalyst), and N,N-dimethylformamide (32.5 g) were mixed and stirred thoroughly to yield a solution. Next, the liquid precursor was injected into a reacting cell made of two pieces of glass with a silicone rubber spacer, followed by UV light exposure (UV TaoYuan, 365 nm) for 1 min. Finally, the solvent was evaporated by leaving the composite film (0.26 mm thick) in a fume hood at room temperature for 24 h. The preparation of OH0, OH10, and OH50 followed the same procedure except that the amount of 4-hydroxybutyl acrylate was adjusted accordingly.

**Preparation of the composites**. For P29, phenoxyethanol (6.5 g, sintering aid) and silica nanoparticles (7.8 g) were mixed and dispersed into the precursor solution of OH30. The curing and solvent removal procedures were the same as those of the above-mentioned neat polymer processing. With the same procedure, adjusting the loading of silica particles allowed preparation of P0, P17, and P38.

**Origami process**. A polymer composite sheet was first cut into a target shape by a laser cutter (Trotec Speedy 100R with a $CO_2$ laser of 10,640 nm). The sample was either deformed into 3D at room temperature (physical plasticity) or deformed at 130 °C for 2 h (chemical plasticity). For the physical plasticity process, the sample sheet was deformed manually at room temperature much like typical paper origami. For the chemical plasticity process, two thin polydimethylsiloxane (PDMS) film were stuck on the sample sheet to prevent surface sticking. The PDMS-protected sample was deformed manually at room temperature, fixed by clips, and annealed at 130 °C for 2 h. After that, the PDMS films were peeled off to complete the process.

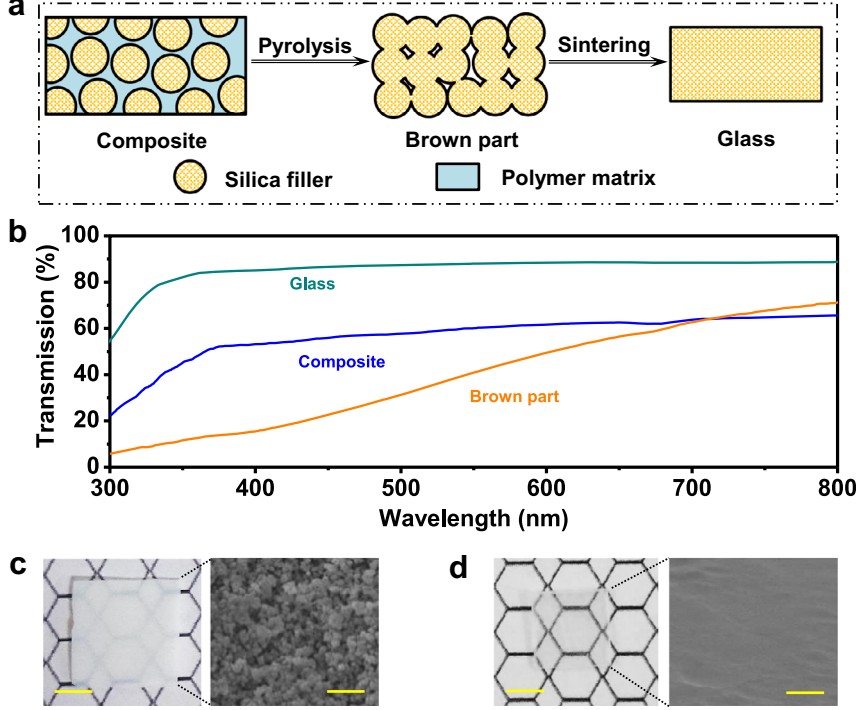

**Fig. 3 Fabrications of the glass. a** Schematic illustration of the pyrolysis and sintering steps for the composite. The green area is the polymer scaffold and the yellow area is the silica. **b** The UV-vis transmission spectrum of the composite green body, the brown part, and the glass. **c** The macroscopic photos and microscopic SEM images of the brown part. **d** The macroscopic photos and microscopic SEM images of the transparent glass. The scale bars of the photos are 1 cm and SEM images are 300 nm.

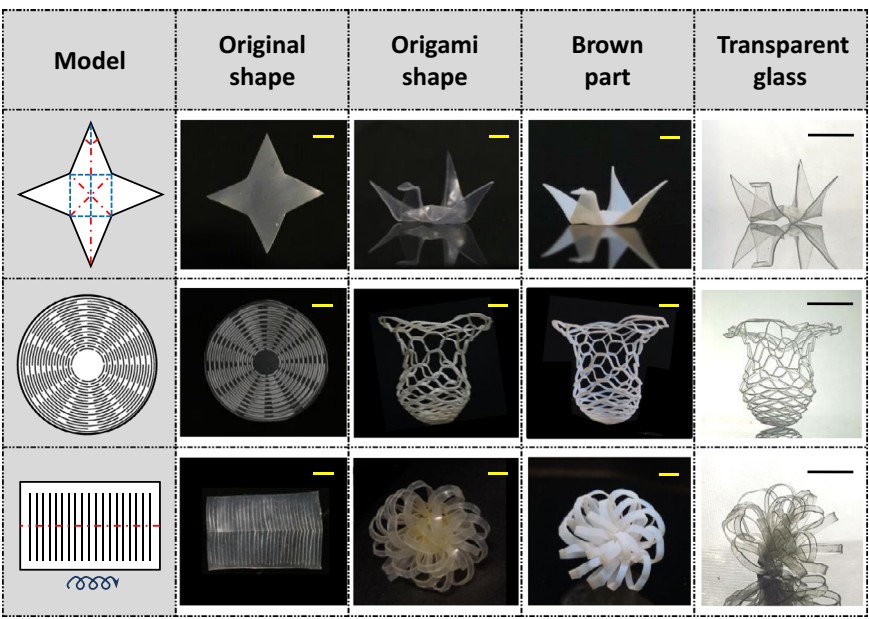

**Fig. 4 Fabrication of transparent origami glass.** In the models, the black lines, the red dashed lines, and the blue dashed lines represent cutting, mountain folding, and valley folding, respectively. The blue twisting arrow represents curling. All scale bars: 1 cm.

**Pyrolysis and vacuum sintering**. The deformed composite film was pyrolyzed in a KSL-1700X muffle furnace (HF-kejing) and then sintering in a GSL-1700X vacuum tube furnace (HF-kejing) at a reduced pressure of 1 kPa. The pyrolysis procedure: (i) 5 °C/min, from room temperature to 170 °C, kept for 3 h; (ii) 0.5 °C/min, from 170 to 300 °C, kept for 5 h; (iii) 0.5 °C/min, from 300 to 600 °C, kept for 5 h; (iv) furnace cooling. The sintering procedure: (i) 10 °C/min, from room temperature to 1050 °C; (ii) 1 °C/min, from 1050 to 1300 °C; (iii) 5 °C/min, from 1300 to 1050 °C, kept for 2 h; (iv) furnace cooling. Illustrated heating procedure is shown in Supplementary Fig. 4.

**Mechanical and thermomechanical characterizations**. Young's moduli and strains at break were measured using a Zwick/Roell Z005 machine in a tensile mode at room temperature at a tensile speed of 10 mm/min. Isostrain stress relaxation curves were obtained using dynamic thermomechanical analysis (TA Q800 machine) under a "strain rate" mode. The tensile shape retention was measured manually by comparing the sample length before and after the deformation. The bending shape retention was measured manually by comparing the bending angle before and after the deformation. The bending shape retention is defined as $R_{bren} = 100\% \times (\theta/\theta_{load})$, with the $\theta_{load}$ and $\theta$ denoting the applied angle

and the permanently fixed angle (after the stress relaxation and force removal). The $\theta_{load}$ is 180°.

**Surface morphology characterizations**. The SEM images were obtained using an SU-8010 instrument (Hitachi). The EDS images were obtained using an Oxford X-max80 machine. Surface roughness was measured using a VEECO MultiMode instrument.

**Glass characterizations**. XRD was characterized using an X'-pert Powder instrument (Malvern Panalytical). Optical transmission was measured using a UV-vis spectrometer (model U4800 by Hitachi).

## Data availability

The data that support the findings of this study are available from the corresponding author on reasonable request.

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

## Acknowledgements

This work is supported by the National Natural Science Foundation of China (Nos. 52033009, 52003232, and 21625402), China National Postdoctoral Program for Innovative Talents (BX20190294), and China Postdoctoral Science Foundation (2020M681821). We thank Prof. Jianrong Qiu, Dr. Chang Liu, and Mr. Dao Zhang for their discussion on glass sintering, Mrs. Li Xu, and Mrs. Na Zheng for their assistance in performing TGA and SEM analyses at State Key Laboratory of Chemical Engineering (Zhejiang University).

## Author contributions

T.X. and N.Z. conceived the concept. Y.X. designed and conducted the experiments. Y.L. performed the experiments. Y.X., N.Z., and T.X. wrote the paper. Y.X. and N.Z. analyzed experimental results. Y.X., N.Z., Q.Z., and T.X. contributed to the discussion.

## Competing interests

The authors declare no competing interests.
