## [Peer Review File · Nature Communications]

REVIEWER COMMENTS

Reviewer #1 (Remarks to the Author):

The manuscript entitled "Transparent Origami Glass" by Xu et al. is a well-written account of a successful, novel, creative study combining the art of origami with dynamic covalent polymer nanocomposites at high filler loading and processing by pyrolysis and sintering. The results of their study are some amazing examples of transparent, complex-shaped, 3D glass. This reviewer would like to congratulate the authors on their accomplishment. This reviewer believes that this manuscript merits publication in Nature Communications after the authors address a small number of minor comments. After minor revision, the editor alone should be able to determine whether the authors adequately addressed the comments given below, and there is no need to send the revised manuscript back to this reviewer for re-review.

Minor Comments

1. Page 2, line 33 – Should "silica polymer composites" be "silica/polymer composites" or "silica-polymer composites"?
2. Page 5, lines 95 to 97 -- Could the authors comment on why the stress relaxation rate has a maximum at OH30? Is this due to T_g effects? Other effects?
3. Page 5, lines 106-107 -- The authors should provide a reference regarding the toughening effect of silica leading to an increase in strain at break.
4. Page 5, line 110 – "rention" should be "retention" and "room temperature physical plasticity" should be "room-temperature physical plasticity".
5. Page 6, line 117 – "rention" should be "retention".
6. The paragraph starting on page 5 and ending on page 6 is quite long, and the reviewer had to do some re-reading. It may be better to paragraph starting with the sentence on page 6, line 118, "To understand the mechanism..."
7. Page 6, line 6 – If "nanocomposites" is correct (and the reviewer believes it is), then the term "nano-filler" used in this line should be "nanofiller".
8. Page 7, line 136 – The authors argue that the glass is "pure silica" after saying that the residual mass of pure polymer network after pyrolysis is "near zero". The term "pure" is absolute. Would the authors not be better off saying "pure or nearly pure silica"?
9. Page 7, line 140 – How do the authors determine 17 nm from the Supplementary Fig. 5? The reviewer thinks that the number may be larger than that after looking at Fig. 5. The authors could provide a brief explanation with Supplementary Fig. 5 to address this issue.
10. Page 7, lines 143 to 144 – Could the authors add a phrase or a sentence to indicate to readers WHY the XRD analysis shows the amorphous nature? Many readers may not be familiar with XRD.
11. Page 8, line 163 – "large scale manufacturing" should be "large-scale manufacturing".
12. Page 10, line 204 – Were the tensile measurements actually done at 25 degrees C or at room temperature?
13. Figure 2a and 2b and Figure 3b – As a general guideline, it is usually a good idea for figures to be designed so that if readers are viewing a black-and-white printout or are color-blind, they can nevertheless easily tell what is being plotted or shown. These figures do not follow that guideline
14. Figure 2d – None of the very small data points shows any error bars. Should there be error bars or should the size of the symbols be larger? Or are the measurements of shape retention super-repeatable and incredibly precise?
15. Supplementary Fig. 1 – A lot of abbreviations are used here that are never employed in the manuscript. In the figure caption, the authors should include the full names and the abbreviations to aid the reader.
16. Supplementary Fig. 2 – It would be useful for the authors to tell the readers the value of the stress at zero time.
17. Supplementary Fig. 5 – It would be better to have the y-axis of this plot be larger than its current micro-size.

Reviewer #2 (Remarks to the Author):

In the present work, the authors have proposed a strategy that allows making 3D transparent glass with origami technique. Two plasticity mechanisms, particle cavitation and dynamic bond exchange, are used in the glass origami technique to maintain the deformed shape during the high temperature pyrolysis step. It's a novel work and meaningful for the manufacturing of 3D transparent glass objects. However, the following problems should be addressed before acceptance for publication.

1. In the material synthesis procedure, what is the role of dibutyltin dilaurate (DBTDL) in the synthesis of polycaprolactone diacrylate (PCLDA) and the dynamic covalent network polymer matrix? What is the role of phenoxyethanol (POE) in the synthesis of polymer matrix OH30 and composite P36?

2. In Fig. 2d, the maximum shape retention of P36 is only 37%. I wonder if it is enough for the origami.

3. In supplementary Fig. 2, the shape retention of OH30 reaches 87% at 130°C for 2 h. Does the composite P36 have the same high shape retention as OH30? The stress relaxation and strain curves of P36 at 130 °C should also be measured.

4. The particle size of SiO₂ used in this work is 50 nm. Is there any relationship between the particle size of SiO₂ and mechanical properties of the composites? What happens if using SiO₂ with different size?

5. The origami is performed at room temperature. Could the deformed shape be maintained when heating to high temperature without applying any load?

6. In most cases, origami belongs to bending deformation. The shape retention under tensile stress might be not accurate enough to characterize the origami ability. I think it is better to add some experiments to measure the shape retention under bending stress.

7. The SiO₂ content of the composite P36 used for origami in this work is only 36%, and the linear shrinkage after sintering reaches 46%. Moreover, thin objects are prone to unpredictable deformation during the pyrolysis and sintering. How to ensure that the shape will not be deformed during the heating procedure?

8. According to Supplementary Fig. 4, the residual mass of P36 after heating at 600 °C is about 31-32%, which is inconsistent with the 36% stated in the manuscript (Line 134, Page 6). Could the authors explain it?

9. Please add more discussions about the heating procedure since the pyrolysis and sintering are critical procedures for the formation of transparent glass. Could the authors explain the heating curve in Supplementary Fig. 3? This curve is very different from the conventional sintering curve. Why is the heating rate so fast from 1050 °C to 1300 °C? Why is the cooling rate so fast from 1050 °C to 400 °C and what means is it achieved by? Why is the cooling rate so slow below 400 °C. Why is the pyrolysis proceeded in the air and the sintering in vacuum? Is it better if the pyrolysis in a vacuum?

10. The cracks affect the mechanical strength significantly. Does the 3D glass have any cracks after sintering and how to avoid the cracks? How about the mechanical properties of the transparent glass after sintering? The hardness and bending strength should be tested.

11. What is the thickness of the composite film in this work? What is the maximum thickness of film that could be folded by this origami technique? Are there any wrinkles during the photocuring when producing a relative thick film? Under the current experimental conditions, what curing depth could reach? Is it possible to directly print the designed flat patterns by DLP or SLA 3D printing without laser cutting?

12. What are the factors that affect the transparency of glass in this work? Is there any way to further increase its transmittance?

Reviewer #3 (Remarks to the Author):

Glass is a very brittle material (except at high temperatures) and so forming it into complex shapes is difficult without fracturing or producing areas of high stress and making it extremely fragile. In this manuscript, the authors present a glass forming composite that can be folded like paper and then later turned into glass, allowing complex 3D glass structures to be produced with origami techniques.

The authors have made an exciting contribution in the field of origami engineering by expanding the materials that can be used. The following should be addressed prior to publication.

- 1) In the introduction, the authors mention that origami with glass is difficult owing to its brittle nature. While true, there have been some progress reports in laser forming that allows pre-formed glass sheets to be bent using a laser such as Laser Bending of Brittle Materials by Wu et al. (2010) <https://doi.org/10.1016/j.optlaseng.2009.09.009>
- 2) In the methods section, every chemical was identified except for the UV curing initiator and the transesterification catalyst. As these were not varied during these studies, please list them.
- 3) Why do the authors use a "mass basis" for their investigation of the ratio of the polycaprolactone diacrylate and 4-hydroxybutyl acrylate on chemical plasticity when chemistry occurs on a molar basis?
- 4) In figure 2, please use solid colors for the bar graphs as the diagonal lines are dizzying and obscuring the error bars. Please make the error bars in black to make them easier to see (2b). On error bars, why is Figure 2b the only data with error bars.
- 5) The writing is unclear in places and often uses the phrase "origami technique" to describe the plasticity mechanism rather than how the folding is accomplished which is done manually in this work.
- 6) The authors claim that their technique is suitable for large scale manufacturing, but they should have a reference to show that structures made using origami techniques have been manufactured on a large scale, which to the knowledge of this reviewer, has not been.
- 7) The methods section needs substantial rewrites in order to make this protocol robust enough for others to reproduce their experiments.
 - a. How was the BHT stabilizer removed from the 2-isocyanatoethyl acrylate, if at all, prior to synthesis?
 - b. Is there a reason the molar quantities are not the same? (~30 mmol of -OH from the diol but ~44 mmol of the 2-isocyanatoethyl acrylate)
 - c. The authors claimed to have prepared pure polycaprolactone diacrylate. There is no description in the methods section of what analytical methods they used to characterize conversion of their product. The authors do present a grainy NMR (Bruker is the manufacturer of the NMR) in supplementary figure 1, but it is insufficient to assess the conversion of the diol to diacrylate. This experiment should be redone with an internal standard (such as 1,3,5-trioxane e.g.) that has a sharp proton NMR peak that does not overlap with any others in the sample and does not degrade during synthesis. The integrated values of the NMR peaks can then be used to confirm the conversion. As it stands, the NMR only

confirms that there are some acrylate groups in the NMR sample. A complementary technique such as MALDI-TOF mass spectrometry would also help clarify this (provided the polydispersity is low).

d. It is a bit odd that the 4-hydroxybutyl acrylate did not have an inhibitor in it like the 2-isocyanatoethyl acrylate. Please confirm and if there was an inhibitor or not, and if there was one, how was it removed?

e. The Trotec speedy series offer multiple different laser wavelengths. Please specify what type of laser was used for the laser cutting.

f. The "origami process" section needs to be greatly expanded. As written, it suggests that these glass forming sheets are self-folding, which is a valuable contribution, but the conclusions state that the folding was done manually. Given the folding was done manually, how was the origami folding accomplished at 130 °C?

Authors' response to reviewers

We thank the reviewers for their constructive comments. This manuscript has been revised accordingly with the changes in the main text highlighted in red. Below are the point-by-point responses to the review comments with our responses highlighted in bold.

Reviewer #1:

The manuscript entitled "Transparent Origami Glass" by Xu et al. is a well-written account of a successful, novel, creative study combining the art of origami with dynamic covalent polymer nanocomposites at high filler loading and processing by pyrolysis and sintering. The results of their study are some amazing examples of transparent, complex-shaped, 3D glass. This reviewer would like to congratulate the authors on their accomplishment. This reviewer believes that this manuscript merits publication in Nature Communications after the authors address a small number of minor comments. After minor revision, the editor alone should be able to determine whether the authors adequately addressed the comments given below, and there is no need to send the revised manuscript back to this reviewer for re-review.

Minor Comments

1. Page 2, line 33 – Should "silica polymer composites" be "silica/polymer composites" or "silica-polymer composites"?

Author Response: "silica polymer composites" has been changed into "silica-polymer composites".

2. Page 5, lines 95 to 97 -- Could the authors comment on why the stress relaxation rate has a maximum at OH30? Is this due to Tg effects? Other effects?

Author Response: The explanation has been added as follows, "This is most likely due to the fact that the reduction in the number of ester bonds overtakes the increase of the hydroxyl groups."

3. Page 5, lines 106-107 -- The authors should provide a reference regarding the toughening effect of silica leading to an increase in strain at break.

Author Response: A reference (Wu, C. L., Zhang, M. Q., Rong, M. Z., Lehmann, B. & Friedrich, K. Deformation characteristics of nano-SiO₂ filled polypropylene composites. *Polym. Polym. Compos.* 2003, 11, 559-562) has been cited as Ref. 24.

4. Page 5, line 110 – “rention” should be “retention” and “room temperature physical plasticity” should be “room-temperature physical plasticity”.

Author Response: Revised accordingly.

5. Page 6, line 117 – “rention” should be “retention”.

Author Response: Revised accordingly.

6. The paragraph starting on page 5 and ending on page 6 is quite long, and the reviewer had to do some re-reading. It may be better to paragraph starting with the sentence on page 6, line 118, “To understand the mechanism...”

Author Response: Thanks for your suggestion. For better readability, the mentioned paragraph has been divided into two paragraphs.

7. Page 6, line 6 – If “nanocomposites” is correct (and the reviewer believes it is), then the term “nano-filler” used in this line should be “nanofiller”.

Author Response: Change made accordingly.

8. Page 7, line 136 – The authors argue that the glass is “pure silica” after saying that the residual mass of pure polymer network after pyrolysis is “near zero”. The term “pure” is absolute. Would the authors not be better off saying “pure or nearly pure silica”?

Author Response: “pure silica” has been changed to “silica”.

9. Page 7, line 140 – How do the authors determine 17 nm from the Supplementary Fig. 5? The reviewer thinks that the number may be larger than that after looking at Fig. 5. The authors could provide a brief explanation with Supplementary Fig. 5 to address this issue.

Author Response: A brief explanation has been added in the figure caption of the Supplementary Fig. 6 (original Supplementary Fig. 5) as follows, “*The surface roughness R_a , defined as the arithmetical mean of the absolute value of the deviation of the surface profile, is within 17 nm.*”

10. Page 7, lines 143 to 144 – Could the authors add a phrase or a sentence to indicate to readers WHY the XRD analysis shows the amorphous nature? Many readers may not be familiar with XRD.

Author Response: The explanation has been added in the figure caption of Supplementary Fig. 7 (original Supplementary Fig. 6) as follows, “*The broad diffraction peak indicates the amorphous nature of the sintered glass.*”

11. Page 8, line 163 – “large scale manufacturing” should be “large-scale manufacturing”.

Author Response: Revised accordingly.

12. Page 10, line 204 – Were the tensile measurements actually done at 25 degrees C or at room temperature?

Author Response: The tensile measurements were all performed at room temperature. Thanks for your kind advice, and the mentioned text “25 °C” has been replaced with “room temperature”.

13. Figure 2a and 2b and Figure 3b – As a general guideline, it is usually a good idea for figures to be designed so that if readers are viewing a black-and-white printout or are color-blind, they can nevertheless easily tell what is being plotted or shown. These figures do not follow that guideline.

Author Response: The bar graphs in Figure 2b and 2c have been revised to solid colors and error bars have been changed into black.

14. Figure 2d – None of the very small data points shows any error bars. Should there be error bars or should the size of the symbols be larger? Or are the measurements of shape retention super-repeatable and incredibly precise?

Author Response: Additional tests have been performed and error bars have been added in Figures 2c and 2d.

15. Supplementary Fig. 1 – A lot of abbreviations are used here that are never employed in the manuscript. In the figure caption, the authors should include the full names and the abbreviations to aid the reader.

Author Response: The full names and their corresponding abbreviations have been added in the caption of Supplementary Fig. 1.

16. Supplementary Fig. 2 – It would be useful for the authors to tell the readers the value of the stress at zero time.

Author Response: The description has been added in the caption of Supplementary Figure 2 as follows “*The absolute stress at time zero was 0.18 MPa.*”

17. Supplementary Fig. 5 – It would be better to have the y-axis of this plot be larger than its current micro-size.

Author Response: Supplementary Fig. 6 (original Supplementary Fig. 5) has been revised accordingly.

Reviewer #2:

In the present work, the authors have proposed a strategy that allows making 3D transparent glass with origami technique. Two plasticity mechanisms, particle cavitation and dynamic bond exchange, are used in the glass origami technique to maintain the deformed shape during the high temperature pyrolysis step. It's a novel work and meaningful for the manufacturing of 3D transparent glass objects. However, the following problems should be addressed before acceptance for publication.

1. In the material synthesis procedure, what is the role of dibutyltin dilaurate (DBTDL) in the synthesis of polycaprolactone diacrylate (PCLDA) and the dynamic covalent network polymer matrix? What is the role of phenoxyethanol (POE) in the synthesis of polymer matrix OH30 and composite P36?

Author Response: DBTDL is the reaction catalyst between the hydroxyl groups and isocyanate groups in the synthesis of PCLDA. It also works as the transesterification catalyst in the dynamic ester bond exchange. POE was added in the composite P36 as a sintering aid. There was no POE adding in OH30. The description has been added in the Methods section (page 9, 10) highlighted in red.

2. In Fig. 2d, the maximum shape retention of P36 is only 37%. I wonder if it is enough for the origami.

Author Response: 37% is the physical shape retention, while the chemical shape retention is over 90%. Actual paper origami does not have very high shape retention, but it does not impact the process. Our overall results shown in Figure 4 prove that the shape retention does not impact the origami process. No revision has been made.

3. In supplementary Fig. 2, the shape retention of OH30 reaches 87% at 130 °C for 2 h. Does the composite P36 have the same high shape retention as OH30? The stress relaxation and strain curves of P36 at 130 °C should also be measured.

Author Response: The shape retention of P36 (now renamed as P29, see response to comment #8) is slightly higher than that of OH30 because of the synergism of the physical and chemical plasticity. The tensile shape retention has been measured and details have been added in the main text as follows, "The tensile shape retention of P29 under chemical plasticity is $91\pm 2\%$ (Supplementary Fig. 3a)". The measurement has been added in the Methods section highlighted in red.

4. The particle size of SiO₂ used in this work is 50 nm. Is there any relationship between the particle size of SiO₂ and mechanical properties of the composites? What happens if using SiO₂ with different size?

Author Response: The particle size affects the mechanical properties of the composites, which is well known in the literature (e.g. *Macromolecules* 2016, 49,

7077-7097). However, we consider this is beyond the focus of the current work. No revision has been made.

5. The origami is performed at room temperature. Could the deformed shape be maintained when heating to high temperature without applying any load?

Author Response: After the origami process via physical plasticity at room temperature, the relative position between nanoparticle and polymer matrix has been permanently locked. The deformed shape is maintained when heating to high temperature without applying any load. No revision has been made.

6. In most cases, origami belongs to bending deformation. The shape retention under tensile stress might be not accurate enough to characterize the origami ability. I think it is better to add some experiments to measure the shape retention under bending stress.

Author Response: The bending shape retention has been measured and the results have been added in the main text (page 6) as follows, *“By comparison, the bending shape retention after physical plasticity is 55%±3%, while the bending shape retention after chemical plasticity is 92%±3% (Supplementary Fig. 3b)”*.

7. The SiO₂ content of the composite P36 used for origami in this work is only 36%, and the linear shrinkage after sintering reaches 46%. Moreover, thin objects are prone to unpredictable deformation during the pyrolysis and sintering. How to ensure that the shape will not be deformed during the heating procedure?

Author Response: We did not experience this issue for all demonstrations except the vase in Figure 4. The following statement has been added on page 8. *“In particular, no unpredictable deformation was observed during the pyrolysis and sintering process except the vase shape that requires an extra mechanical support.”*

8. According to Supplementary Fig. 4, the residual mass of P36 after heating at 600 °C is about 31-32%, which is inconsistent with the 36% stated in the manuscript (Line 134, Page 6). Could the authors explain it?

Author Response: This discrepancy is due to the naming. The composite sample P36, for instance, represent silica percentage (36%) over the total weight of silica and polymer. With this naming, the weight of the small molecular sintering aid (POE) is not counted. In TGA (original Supplementary Fig. 4), the residual mass percentage corresponds to the weight percentage of silica over the total weight including POE. To avoid this confusion, we have renamed the composite sample based on the weight percentage of silica over the total weight. The names of the composites have been changed accordingly. Note that these changes do not affect any of the discussions or conclusions.

9. Please add more discussions about the heating procedure since the pyrolysis and sintering are critical procedures for the formation of transparent glass. Could the authors explain the heating curve in Supplementary Fig. 3? This curve is very

different from the conventional sintering curve. Why is the heating rate so fast from 1050 °C to 1300 °C? Why is the cooling rate so fast from 1050 °C to 400 °C and what means is it achieved by? Why is the cooling rate so slow below 400 °C. Why is the pyrolysis proceeded in the air and the sintering in vacuum? Is it better if the pyrolysis in a vacuum?

Author Response: In the sintering step, the heating from 1050 °C to 1300 °C was designed to be rapid in order to utilize the kinetic difference between the surface migration and the grain growth to promote transparency of the resulting glass. The cooling from 1050 °C to ambient temperature was not purposely controlled. It was natural furnace cooling. The supplementary figure caption has been revised accordingly. The pyrolysis was conducted in the air in order to promote oxidation to avoid carbonization of the polymer. By comparison, sintering was performed under vacuum to ensure densification, which key to transparency of the final glass. This discussion is also included in the caption of Supplementary Figure 4 (original Supplementary Figure 3).

10. The cracks affect the mechanical strength significantly. Does the 3D glass have any cracks after sintering and how to avoid the cracks? How about the mechanical properties of the transparent glass after sintering? The hardness and bending strength should be tested.

Author Response: Cracks are developed in the physical plasticity process. During pyrolysis, the removal of the polymer phase leaves voids in the brown part. These voids are removed during the densification process sintering step. The cracks are also removed in the same process. Nevertheless, we have measured the hardness and bending strength. The following description has been added in main text (Page 7) as follows, “*The bending strength and hardness of the obtained glass, calculated from the mechanical curves in Supplementary Fig. 8, are 88.2 ± 5.3 MPa and 8.2 ± 0.3 GPa, respectively.*”.

11. What is the thickness of the composite film in this work? What is the maximum thickness of film that could be folded by this origami technique? Are there any wrinkles during the photocuring when producing a relative thick film? Under the current experimental conditions, what curing depth could reach? Is it possible to directly print the designed flat patterns by DLP or SLA 3D printing without laser cutting?

Author Response: The composite thickness of 0.26 mm is added in the Method section on Page 10.

We did not investigate quantitatively how the thickness impacts the process, which we consider this outside the scope of the current study. However, we did attempt to make a thicker film (around 0.5 mm). In that experiment, we did not observe any visible wrinkles.

In an attempt to investigate the maximum curing step, we discovered that the film can be cured even at 9 cm thickness, way beyond what light attenuation would lead one to believe. Origami, however, is limited to thin film folding. That

the film can be cured at this extreme thickness is intriguing enough that we will devote a separate study on this in the future.

It is possible to use DLP to directly print the flat pattern, instead of laser cutting. The following statement is added on Page 8. *“Alternatively, it is also possible to use digital projector light to directly produce the flat cut pattern if a visible light initiator is employed.”*

12. What are the factors that affect the transparency of glass in this work? Is there any way to further increase its transmittance?

Author Response: The following statement is added on Page 7. *“Increasing the transmission beyond 90% is possible by further optimization of the heating procedure and employing higher vacuum during sintering.*

Reviewer #3:

Glass is a very brittle material (except at high temperatures) and so forming it into complex shapes is difficult without fracturing or producing areas of high stress and making it extremely fragile. In this manuscript, the authors present a glass forming composite that can be folded like paper and then later turned into glass, allowing complex 3D glass structures to be produced with origami techniques.

The authors have made an exciting contribution in the field of origami engineering by expanding the materials that can be used. The following should be addressed prior to publication.

1) In the introduction, the authors mention that origami with glass is difficult owing to its brittle nature. While true, there have been some progress reports in laser forming that allows pre-formed glass sheets to be bent using a laser such as Laser Bending of Brittle Materials by Wu et al. (2010) <https://doi.org/10.1016/j.optlaseng.2009.09.009>

Author Response: This paper has now been cited as reference 9 and related description has been added as follows, “Localized laser processing allows glass shaping, but the method is limited to simple geometric manipulation such as bending⁹.”

2) In the methods section, every chemical was identified except for the UV curing initiator and the transesterification catalyst. As these were not varied during these studies, please list them.

Author Response: Clarification has been added on Page 9 and 10, stating that the Irgacure 2959 is the UV curing initiator and the dibutyltin dilaurate is the transesterification catalyst.

3) Why do the authors use a “mass basis” for their investigation of the ratio of the polycaprolactone diacrylate and 4-hydroxybutyl acrylate on chemical plasticity when chemistry occurs on a molar basis?

Author Response: The molar basis is more accurate in the molecular chemistry and the mass basis is more convenient for experimental operations in the material production. They can be converted easily. No revision is made here.

4) In figure 2, please use solid colors for the bar graphs as the diagonal lines are dizzying and obscuring the error bars. Please make the error bars in black to make them easier to see (2b). On error bars, why is Figure 2b the only data with error bars.

Author Response: The graphs in Figure 2b and 2c have been revised accordingly. Error bars are added in Figures 2c and 2d.

5) The writing is unclear in places and often uses the phrase “origami technique” to describe the plasticity mechanism rather than how the folding is accomplished which is done manually in this work.

Author Response: The following sentence has been added on Page 3, “*which involves manual folding at room temperature much like actual paper origami.*”

6) The authors claim that their technique is suitable for large scale manufacturing, but they should have a reference to show that structures made using origami techniques have been manufactured on a large scale, which to the knowledge of this reviewer, has not been.

Author Response: We cannot find an appropriate reference for this. By stating that our technique is suitable for large-scale manufacturing, all we mean is that the individual steps involved including laser cutting, origami folding at room temperature, chemical plasticity, and high temperature pyrolysis can be potentially scaled up without any difficulty. No revision is made.

7) The methods section needs substantial rewrites in order to make this protocol robust enough for others to reproduce their experiments.

a. How was the BHT stabilizer removed from the 2-isocyanatoethyl acrylate, if at all, prior to synthesis?

Author Response: As the materials section written, the 2-isocyanatoethyl acrylate was used as received, and the BHT stabilizer was not removed. The description has been added as follows, “*All the chemicals were used as received.*”

b. Is there a reason the molar quantities are not the same? (~30 mmol of -OH from the diol but ~44 mmol of the 2-isocyanatoethyl acrylate)

Author Response: The following clarification has been added on Page 9. “We note that 2-isocyanatoethyl acrylate was used in excess to ensure high conversion of the hydroxyl groups and the unreacted 2-isocyanatoethyl acrylate monomer was removed in the subsequent purification process.”

c. The authors claimed to have prepared pure polycaprolactone diacrylate. There is no description in the methods section of what analytical methods they used to characterize conversion of their product. The authors do present a grainy NMR (Bruker is the manufacturer of the NMR) in supplementary figure 1, but it is insufficient to assess the conversion of the diol to diacrylate. This experiment should be redone with an internal standard (such as 1,3,5-trioxane e.g.) that has a sharp proton NMR peak that does not overlap with any others in the sample and does not degrade during synthesis. The integrated values of the NMR peaks can then be used to confirm the conversion. As it stands, the NMR only confirms that there are some acrylate groups in the NMR sample. A complementary technique such as MALDI-TOF mass spectrometry would also help clarify this (provided the polydispersity is low).

Author Response: Thanks for the comment. We did not claim to have prepared pure polycaprolactone diacrylate, which is unrealistic given that this is a reaction between the polymeric chain ends and a small molecule (2-isocyanatoethyl acrylate).

NMR analysis was used to calculate the chain end conversion from hydroxyl groups to acrylates and TMS was used as the internal standard during the analysis. As can be seen in Supplementary Figure 1b, the peaks corresponding to the acrylate in the synthesized polycaprolactone diacrylate are shifted comparing to those in the 2-isocyanatoethyl acrylate. This comparison verifies that the acrylate is indeed grafted onto the polycaprolactone chain. To calculate the hydroxyl to acrylate conversion, we used the following equation $\psi = A_a N / A_f * 100\%$, with A_a , A_f , and N being the area of the defined peak a, the area of the defined peak f, and the number of the repeating units in polycaprolactone, respectively. Based on the above analysis, the diol to diacrylate conversion is calculated as 90.5%. The related discussion and addition can be found as the caption of Supplementary Fig 1.

d. It is a bit odd that the 4-hydroxybutyl acrylate did not have an inhibitor in it like the 2-isocyanatoethyl acrylate. Please confirm and if there was an inhibitor or not, and if there was one, how was it removed?

Author Response: Thanks for the careful reading. This detail has been added in the materials section as follows, “(stabilized with MEHQ)”. Like other chemicals, the 4-hydroxybutyl acrylate was used as received without MEHQ stabilizer removal.

e. The Trotec speedy series offer multiple different laser wavelengths. Please specify what type of laser was used for the laser cutting.

Author Response: This detail has been added in the methods section as follows, “with a CO₂ laser of 10640 nm”.

f. The “origami process” section needs to be greatly expanded. As written, it suggests that these glass forming sheets are self-folding, which is a valuable contribution, but the conclusions state that the folding was done manually. Given the folding was done manually, how was the origami folding accomplished at 130 °C?

Author Response: Thanks for the advice. The following details have been added in the “origami process” section on Page 10 as follows, “For the physical plasticity process, the sample sheet was deformed manually at room temperature much like typical paper origami. For the chemical plasticity process, two thin polydimethylsiloxane (PDMS) film were stuck on the sample sheet to prevent surface sticking. The PDMS-protected sample was deformed manually at room temperature, fixed by clips, and annealed at 130 °C for 2 hours. After that, the PDMS films were peeled off to complete the process”.

REVIEWERS' COMMENTS

Reviewer #2 (Remarks to the Author):

All the comments have been well addressed by the authors. The manuscript is recommended for the publication in Nature Communications.

Reviewer #3 (Remarks to the Author):

Overall, I am happy with how the authors responded to my concerns on the technical aspect of the paper. I think the paper would be improved by adding a paragraph at the end of the introduction clearly stating the claims of the paper. That would be an issue of style that I would leave to the authors to decide.

Authors' response to reviewers

We thank all reviewers for their constructive comments to improve this manuscript.

Reviewer #1

No review comments.

Reviewer #2

All the comments have been well addressed by the authors. The manuscript is recommended for the publication in Nature Communications.

Author Response: Thanks for your review and helpful comments which enrich this manuscript.

Reviewer #3

Overall, I am happy with how the authors responded to my concerns on the technical aspect of the paper. I think the paper would be improved by adding a paragraph at the end of the introduction clearly stating the claims of the paper. That would be an issue of style that I would leave to the authors to decide.

Author Response: Thanks for your kind and useful advices during the review process. We think the existing claims at the end of the Introduction paragraph is clear enough, which is the origami-shaping of transparent glass. No revision has been made.